# OFFLINE META LEARNING OF EXPLORATION

## ABSTRACT

Consider the following problem: given the complete training histories of $N$ conventional RL agents, trained on $N$ different tasks, design a meta-agent that can quickly maximize reward in a new, unseen task from the same task distribution. In particular, while each conventional RL agent explored and exploited its own different task, the meta-agent must identify regularities in the data that lead to effective exploration/exploitation in the unseen task. This meta-learning problem is an instance of a setting we term Offline Meta Reinforcement Learning (OMRL). To solve our challenge, we take a Bayesian RL (BRL) view, and seek to learn a Bayes-optimal policy from the offline data. We extend the recently proposed VariBAD BRL algorithm to the off-policy setting, and demonstrate learning of approximately Bayes-optimal exploration strategies from offline data using deep neural networks. For the particular problem described above, our method learns effective exploration behavior that is qualitatively different from the exploration used by any RL agent in the data. Furthermore, we find that when applied to the online meta-RL setting (agent simultaneously collects data and improves its meta-RL policy), our method is significantly more sample efficient than the state-of-the-art VariBAD.

## 1 INTRODUCTION

A central question in reinforcement learning (RL) is how to learn quickly (i.e., with few samples) in a new environment. Meta-RL addresses this issue by assuming a distribution over possible environments, and having access to a large set of environments from this distribution during training (Duan et al., 2016; Finn et al., 2017). Intuitively, the meta-RL agent can learn regularities in the environments, which allow quick learning in any environment that shares a similar structure. Indeed, recent work demonstrated this by training memory-based controllers that can 'identify' the domain (Duan et al., 2016; Rakelly et al., 2019; Humplik et al., 2019), or by learning a parameter initialization that can lead to good performance with only a few gradient updates (Finn et al., 2017).

Another approach to quick RL is Bayesian RL (Ghavamzadeh et al., 2016, BRL). In BRL, the environment parameters are treated as unobserved variables, with a known prior distribution. Consequentially, the standard problem of maximizing expected returns (taken with respect to the posterior distribution) *explicitly accounts for the environment uncertainty*, and its solution is a *Bayes-optimal* policy, wherein actions optimally balance exploration and exploitation. Recently, Zintgraf et al. (2020) showed that meta-RL is in fact an instance of BRL, where the meta-RL environment distribution is simply the BRL prior. Furthermore, a Bayes-optimal policy can be trained using standard policy gradient methods, simply by adding to the state the posterior belief over the environment parameters. The VariBAD algorithm (Zintgraf et al., 2020) is an implementation of this approach that uses a variational autoencoder (VAE) for parameter estimation and deep neural network policies.

Most meta-RL studies, including VariBAD, have focused on the *online* setting, where, during training, the meta-RL policy is continually updated using data collected from running it in the training environments. In domains where data collection is expensive, such as robotics and healthcare to name a few, online training is a limiting factor. For standard RL, offline (a.k.a. batch) RL mitigates this problem by learning from data collected beforehand by an arbitrary policy (Ernst et al., 2005; Levine et al., 2020). In this work we investigate the *offline approach to meta-RL* (OMRL).

It is well known that any offline RL approach is heavily influenced by the data collection policy. To ground our investigation, we focus on the following practical setting: we assume that data has been collected by running standard RL agents on a set of environments from the environment distribution.

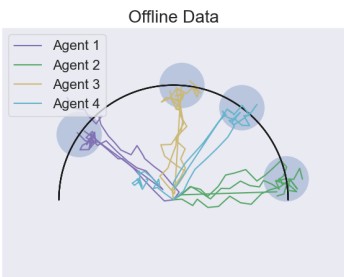 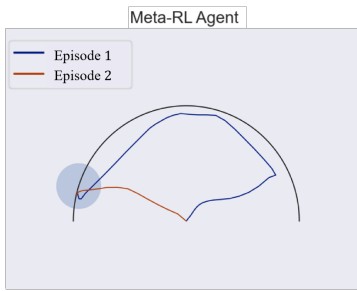

Figure 1: Illustration of offline meta-RL: the task is to navigate to a goal position that can be anywhere on the semi-circle. The reward is sparse (light-blue), and the offline data (left) contains training histories of conventional RL agents trained to find individual goals. The meta-RL agent (right) needs to find a policy that quickly finds the unknown goal, here, by searching across the semi-circle. Note that this search behavior is completely different than the dominant behaviors in the data.

Importantly - we do not allow any modification to the RL algorithms used for data collection, and the meta-RL learner must make use of data that was not specifically collected for the meta-RL task. Nevertheless, we hypothesize that regularities between the training domains can still be learned, to provide faster learning in new environments. Figure 1 illustrates our problem: in this navigation task, each RL agent in the data learned to find its own goal, and converged to a behavior that quickly navigates toward it. The meta-RL agent, on the other hand, needs to learn a completely different behavior that effectively searches for the unknown goal position.

Our key idea to solving OMRL is an off-policy variant of the VariBAD algorithm, based on replacing the on-policy policy gradient optimization in VariBAD with an off-policy Q-learning based method. This, however, requires some care, as Q-learning applies to states of fully observed systems. We show that the VariBAD approach of augmenting states with the belief in the data applies to the off-policy setting as well, leading to an effective algorithm we term Off-Policy VariBAD. The offline setting, however, brings about another challenge – when the agent visits different parts of the state space in different environments, it becomes challenging to obtain an accurate belief estimate, a problem we term MDP ambiguity. When the ambiguity is due to differences in the reward between the environments, we propose a simple solution, based on a reward relabelling trick that significantly improves the performance of the VariBAD VAE trained on offline data. Our experimental results show that our method can learn effective exploration policies from offline data on both discrete and continuous control problems.

Our main contributions are as follows. To our knowledge, this is the first study of meta learning exploration in the offline setting. We provide the necessary theory to extend VariBAD to off-policy RL. We show that a key difficulty in OMRL is MDP ambiguity, and propose an effective solution for the case where tasks differ in their rewards. We show non-trivial empirical results that demonstrate significantly better exploration than meta-RL methods based on Thompson sampling such as PEARL (Rakelly et al., 2019), even when these methods are allowed to train online. Finally, our method can also be applied in the online setting, and demonstrates significantly improved sample efficiency compared to conventional VariBAD.

## 2 BACKGROUND

Our work leverages ideas from meta-RL, BRL and the VariBAD algorithm, as we now recapitulate.

**Meta-RL:** In meta-RL, a distribution over tasks is assumed. A task $\mathcal{T}_i$ is described by a Markov Decision Process (MDP, Bertsekas, 1995) $\mathcal{M}_i = (\mathcal{S}, \mathcal{A}, \mathcal{R}_i, \mathcal{P}_i)$, where the state space $\mathcal{S}$ and the action space $\mathcal{A}$ are shared across tasks, and $\mathcal{R}_i$ and $\mathcal{P}_i$ are task specific reward and transition functions. Thus, we write the task distribution as $p(\mathcal{R}, \mathcal{P})$. For simplicity, we assume throughout that the initial state distribution $P_{init}(s_0)$ is the same for all MDPs. The goal in meta-RL is to train an agent that can quickly maximize reward in new, unseen tasks, drawn from $p(\mathcal{R}, \mathcal{P})$. To do so, the agent must leverage any shared structure among tasks, which can typically be learned from a set of training tasks.

**Bayesian Reinforcement Learning:** The goal in BRL is to find the optimal policy $\pi$ in an MDP, when the transitions and rewards are not known in advance. Similar to meta-RL, we assume a prior

over the MDP parameters $p(\mathcal{R}, \mathcal{P})$, and seek to maximize the expected discounted return,

$$\mathbb{E}_\pi \left[ \sum_{t=0}^\infty \gamma^t r(s_t, a_t) \right], \tag{1}$$

where the expectation is taken with respect to *both the uncertainty in state-action transitions* $s_{t+1} \sim \mathcal{P}(\cdot|s_t, a_t)$, $a_t \sim \pi$, *and the uncertainty in the MDP parameters* $\mathcal{R}, \mathcal{P} \sim p(\mathcal{R}, \mathcal{P})$.[1] A key observation is that this formulation naturally accounts for the exploration/exploitation tradeoff – an optimal agent must plan its actions to reduce uncertainty in the MDP parameters, if such leads to higher rewards.

One way to approach the BRL problem is to model $\mathcal{R}, \mathcal{P}$ as unobserved state variables in a partially observed MDP (POMDP, Cassandra et al., 1994), reducing the problem to solving a particular POMDP instance where the unobserved variables cannot change in time ($\mathcal{P}$ and $\mathcal{R}$ are assumed to be stationary). The *belief* at time $t$, $b_t$, denotes the posterior probability over $\mathcal{R}, \mathcal{P}$ given the history of state transitions and rewards observed until this time $b_t = P(\mathcal{R}, \mathcal{P}|h_{:t})$, where $h_{:t} = \{s_0, a_0, r_1, s_1 \ldots, r_t, s_t\}$ (note that we denote the reward after observing the state and action at time $t$ as $r_{t+1} = r(s_t, a_t)$). The belief can be updated iteratively according to Bayes rule, where $b_0(\mathcal{R}, \mathcal{P}) = p(\mathcal{R}, \mathcal{P})$, and: $b_{t+1}(\mathcal{R}, \mathcal{P}) = P(\mathcal{R}, \mathcal{P}|h_{:t+1}) \propto P(s_{t+1}, r_{t+1}|h_{:t}, \mathcal{R}, \mathcal{P}) b_t(\mathcal{R}, \mathcal{P})$.

Similar to the idea of solving a POMDP by representing it as an MDP over belief states (Cassandra et al., 1994), the state in BRL can be augmented with the belief to result in the Bayes-Adaptive MDP (BAMDP) model (Duff & Barto, 2002). Denote the augmented state $s_t^+ = (s_t, b_t)$ and the augmented state space $\mathcal{S}^+ = \mathcal{S} \times \mathcal{B}$, where $\mathcal{B}$ denotes the belief space. The transitions in the BAMDP are given by: $P^+(s_{t+1}^+|s_t^+, a_t) = \mathbb{E}_{b_t}[\mathcal{P}(s_{t+1}|s_t, a_t)] \delta(b_{t+1} = P(\mathcal{R}, \mathcal{P}|h_{:t+1}))$, and the reward in the BAMDP is the expected reward with respect to the belief: $R^+(s_t^+, a_t) = \mathbb{E}_{b_t}[\mathcal{R}(s_t, a_t)]$. The Bayes-optimal agent seeks to maximize the expected discounted return in the BAMDP, and the optimal solution of the BAMDP gives the optimal BRL policy.

As in standard MDPs, the optimal action-value function in the BAMDP satisfies the Bellman equation:

$$Q(s^+, a) = R^+(s^+, a) + \gamma \sum_{s^{+'} \in \mathcal{S}^+} P^+(s^{+'}|s^+, a) \max_{a'} Q(s^{+'}, a'), \quad \forall s^+ \in \mathcal{S}^+, a \in \mathcal{A}. \tag{2}$$

Computing a Bayes-optimal agent amounts to solving the BAMDP, where the optimal policy is a function of the augmented state. However, for most problems this is intractable, as the augmented state space is continuous and high-dimensional, and the posterior update is also intractable in general.

**The VariBAD Algorithm:** VariBAD (Zintgraf et al., 2020) approximates the Bayes-optimal solution by combining a model for the MDP parameter uncertainty, and an optimization method for the corresponding BAMDP. The MDP parameters are represented by a vector $m \in \mathbb{R}^d$, corresponding to the latent variables in a parametric generative model for the state-reward trajectory distribution conditioned on the actions $P(s_0, r_1, s_1 \ldots, r_H, s_H|a_0, \ldots, a_{H-1}) = \int p_\theta(m) p_\theta(s_0, r_1, s_1 \ldots, r_H, s_H|m, a_0, \ldots, a_{H-1}) dm$. The model parameters $\theta$ are learned by a variational approximation to the maximum likelihood objective, where the variational approximation to the posterior $P(m|s_0, r_1, s_1 \ldots, r_H, s_H, a_0, \ldots, a_{H-1})$ is chosen to have the structure $q_\phi(m|s_0, a_0, r_1, s_1 \ldots, r_t, s_t) = q_\phi(m|h_{:t})$. That is, the approximate posterior is conditioned on the history up to time $t$. The evidence lower bound (ELBO) in this case is $ELBO_t = \mathbb{E}_{m \sim q_\phi(\cdot|h_{:t})}[\log p_\theta(s_0, r_1, s_1 \ldots, r_H, s_H|m, a_0, \ldots, a_{H-1})] - D_{KL}(q_\phi(m|h_{:t})||p_\theta(m))$. The main claim of Zintgraf et al. (2020) is that $q_\phi(m|h_{:t})$ can be taken as an approximation of the belief $b_t$. In practice, $q_\phi(m|h_{:t})$ is represented as a Gaussian distribution $q(m|h_{:t}) = \mathcal{N}(\mu(h_{:t}), \Sigma(h_{:t}))$, where $\mu$ and $\Sigma$ are learned recurrent neural networks.

To approximately solve the BAMDP, Zintgraf et al. (2020) exploit the fact that an optimal BAMDP policy is a function of the state and belief, and therefore consider neural network policies that take the augmented BAMDP state as input $\pi(a_t|s_t, q_\phi(m|h_{:t}))$, where the posterior is practically represented by the distribution parameters $\mu(h_{:t}), \Sigma(h_{:t})$. To train such policies, Zintgraf et al. (2020) maximize the BRL objective,

$$J(\pi) = \mathbb{E}_{\mathcal{R}, \mathcal{P}} \mathbb{E}_\pi \left[ \sum_{t=0}^H \gamma^t r(s_t, a_t) \right], \tag{3}$$

---

[1]For ease of presentation, we consider the infinite horizon discounted return. Our formulation easily extends to the episodic and finite horizon settings.

using policy gradient based methods. The expectation over MDP parameters in (3) is approximated by averaging over training environments, and the RL agent is trained online, alongside the VAE.

# 3 OMRL WITH OFF-POLICY VARIBAD

In this section, we derive an off-policy variant of the VariBAD algorithm, and apply it to the OMRL problem. We begin by describing our OMRL problem setting, and then present our algorithm.

## 3.1 OMRL PROBLEM DEFINITION

We follow the Meta-RL and BRL formulation described above, with a prior distribution over MDP parameters $p(\mathcal{R}, \mathcal{P})$. We are provided training data of an agent interacting with $N$ different MDPs, $\{\mathcal{R}_i, \mathcal{P}_i\}_{i=1}^N$, sampled from the prior. We assume that each interaction is organized as $M$ trajectories of length $H$, $\tau^{i,j} = s_0^{i,j}, a_0^{i,j}, r_1^{i,j}, s_1^{i,j} \ldots, r_H^{i,j}, s_H^{i,j}$, $i \in 1, \ldots, N, j \in 1, \ldots, M$, where the rewards satisfy $r_{t+1}^{i,j} = \mathcal{R}_i(s_t^{i,j}, a_t^{i,j})$, the transitions satisfy $s_{t+1}^{i,j} \sim \mathcal{P}_i(\cdot|s_t^{i,j}, a_t^{i,j})$, and the actions are chosen from an arbitrary data collection policy. To ground our work in a specific context, we further assume that the trajectories are obtained from running a conventional RL agent (i.e., the complete RL training history) in each one of the MDPs, which implicitly specifies the data collection policy. In our experiments, we will later investigate the implications of this assumption. Our goal is to use the data for learning a Bayes-optimal policy, i.e., a policy $\pi$ that maximizes Eq. (1).

## 3.2 OFF-POLICY VARIBAD

The online VariBAD algorithm updates the policy using *trajectories sampled from the current policy*, and thus cannot be applied to our offline setting. Our first step is to modify VariBAD to work off-policy. We start with an observation about the use of the BAMDP formulation in VariBAD, which will motivate our subsequent development.

**Does VariBAD really optimize the BAMDP?** Recall that a BAMDP is in fact a reduction of a POMDP to an MDP over augmented states $s^+ = (s, b)$, and with the rewards and transitions given by $R^+$ and $P^+$. Thus, an optimal Markov policy for the BAMDP exists in the form of $\pi(s^+)$. The VariBAD policy, as described above, similarly takes as input the augmented state, and is thus capable of representing an optimal BAMDP policy. However, *VariBAD's policy optimization in Eq. (3) does not make use of the BAMDP parameters $R^+$ and $P^+$!* While at first this may seem counterintuitive, Eq. (3) is in fact a sound objective for the BAMDP, as we now show[2].

**Proposition 1.** *Let $\tau = s_0, a_0, r_1, s_1 \ldots, r_H, s_H$ denote a random trajectory from a fixed history dependent policy $\pi$, generated according to the following process. First, MDP parameters $\mathcal{R}, \mathcal{P}$ are drawn from the prior $p(\mathcal{R}, \mathcal{P})$. Then, the state trajectory is generated according to $s_0 \sim P_{init}$, $a_t \sim \pi(\cdot|s_0, a_0, r_1, \ldots, s_t)$, $s_{t+1} \sim \mathcal{P}(\cdot|s_t, a_t)$ and $r_{t+1} \sim \mathcal{R}(s_t, a_t)$. Let $b_t$ denote the posterior belief at time $t$, $b_t = P(\mathcal{R}, \mathcal{P}|s_0, a_0, r_1, \ldots, s_t)$. Then*

$$P(s_{t+1}|s_0, a_0, r_1, \ldots, r_t, s_t, a_t) = \mathbb{E}_{\mathcal{R}, \mathcal{P} \sim b_t} \mathcal{P}(s_{t+1}|s_t, a_t), \text{ and,}$$

$$P(r_{t+1}|s_0, a_0, r_1, \ldots, s_t, a_t) = \mathbb{E}_{\mathcal{R}, \mathcal{P} \sim b_t} \mathcal{R}(r_{t+1}|s_t, a_t).$$

For on-policy VariBAD, Proposition 1 shows that the rewards and transitions in each trajectory can be seen as sampled from a distribution that **in expectation** is equal to $R^+$ and $P^+$, and therefore maximizing Eq. 3 is valid.[3] However, off-policy RL does not take as input trajectories, but tuples of the form (state, action, reward, next state), where states and actions can be sampled **from any distribution**. For an arbitrary distribution of augmented states, we must replace the rewards and transitions in our data with $R^+$ and $P^+$, which can be difficult to compute. Fortunately, Proposition 1 shows that when collecting data as described above (by sampling complete trajectories), this is not necessary, as in expectation, the rewards and transitions are correctly sampled from the BAMDP.

---

[2]This result is closely related to the discussion in Ortega et al. (2019), here applied to our particular setting.

[3]To further clarify, if we could calculate $R^+$, replacing all rewards in the trajectories with $R^+$ will result in a lower variance policy update, similar to expected SARSA (Van Seijen et al., 2009).

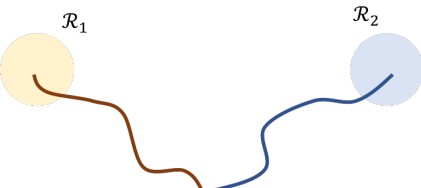

Figure 2: Reward ambiguity: from the two trajectories, it is impossible to know if there are two MDPs with different rewards (blue and yellow circles), or one MDP with rewards at both locations.

Based on Proposition 1, we can use a state augmentation method similar to VariBAD, which we refer to as **state relabelling**. Consider each trajectory in our data $\tau^{i,j} = s_0^{i,j}, a_0^{i,j}, r_1^{i,j}, \ldots, s_H^{i,j}$, as defined above. Recall that the VariBAD VAE encoder provides an estimate of the belief given the state history $q(m|h_{:t}) = \mathcal{N}(\mu(h_{:t}), \Sigma(h_{:t}))$. Thus, we can run the encoder on every partial $t$-length history $\tau_{:t}^{i,j}$ to obtain the belief at each time step. Following the BAMDP formulation, we define the augmented state $s_t^{+,i,j} = (s_t^{i,j}, b_t^{i,j})$, where $b_t^{i,j} = \mu(\tau_{:t}^{i,j}), \Sigma(\tau_{:t}^{i,j})$. We next replace each state in our data $s_t^{i,j}$ with $s_t^{+,i,j}$, effectively transforming the data to as coming from a BAMDP.

After applying state relabelling, any off-policy RL algorithm can be applied to the modified data, for learning a Bayes-optimal policy. In our experiments we used DQN (Mnih et al., 2015) for discrete action domains, and soft actor critic (SAC, Haarnoja et al., 2018) for continuous control.

### 3.3 OFFLINE META-RL AND MDP AMBIGUITY

We next consider the OMRL problem. While in principle, it is possible to simply run off-policy VariBAD on the offline data, we claim that in many problems this may not work well.

The reason is that the VariBAD VAE should reason about the uncertainty of the MDP parameters, which requires to effectively distinguish between the different possible MDPs. However, if trajectories from the different MDPs visit different parts of the state space, it becomes impossible to identify if two trajectories come from different MDPs, or actually come from different parts of the same MDP. This problem, which we term *MDP ambiguity*, is illustrated in Figure 2: there are two MDPs, one with rewards in the blue circle, and the other with rewards in the yellow circle. If the data contains trajectories similar to the ones in the figure, it is impossible to distinguish between having two different MDPs with the indicated rewards, or a single MDP with rewards at both the blue and yellow circles.

Note that MDP ambiguity is special to the offline meta-RL setting, as in online meta-RL, the agent may be driven by the online adapting policy (or guided explicitly) to explore states that reduce its ambiguity. We also emphasize that this problem is not encountered in standard (non-meta) offline RL, as the problem here concerns the *identification of the MDP*, which in standard RL is unique.

We next characterize MDP ambiguity in more detail. Different MDPs can differ from each other either in the reward function, the transition function, or both. The differences can also take place over the whole state space, or only in parts of it – i.e., in some states the MDP rewards or transitions are similar, while on others they are different. Let us denote the set of states where there are such differences as the 'identifying states' of an MDP – if the agent has data on such states obtained from different MDPs, it has the capability to identify which data samples belong to which MDP.
The MDP ambiguity problem arises when for each MDP in the data, the samples that belong to identifying states do not overlap. Since the agent does not know the MDPs in advance, it does not have information to identify from which MDP the data came from (cf. Figure 2).
Problems with sparse rewards, for example, are prone to MDP ambiguity – the identifying states are the sparse set of rewarding states, and by collecting data as described in Section 3.1, each agent mainly visits the rewarding states in its own MDP. It is important to note that there are problems where MDP ambiguity is not an issue. For example, MDPs with different transitions, where the set of rewarding states are similar. In this case, all agents will mainly visit the same states (though by applying different policies), and therefore the MDPs will be identifiable. Another example is when the optimal policy in each MDP visits some set of overlapping states, and these states are identifying. We provide examples of both problems in our experiments. In the following, we separate our discussion to ambiguity due to different rewards, and ambiguity due to different transitions.

**MDP ambiguity due to rewards:** One way to solve MDP ambiguity is to increase the variability of the trajectories in the data, however, this may not be possible if the data has already been collected.

Instead, we propose a simple and effective alternative in case where the MDPs differ only in their reward function. We make the following assumptions:

1. The transition and reward uncertainties are decoupled, i.e., the prior can be written as $p(\mathcal{R}, \mathcal{P}) = p_R(\mathcal{R}) \times p_P(\mathcal{P})$.

2. For each MDP $i$ in the training data, we know the reward function $\mathcal{R}_i$.

These assumptions are largely satisfied in most meta-RL studies to date (e.g., Zintgraf et al. 2020; Li et al. 2020; Finn et al. 2017; Duan et al. 2016). The decoupled uncertainty assumption can be hard to verify in practice. However, we remark that assuming a decoupled prior when the true prior is coupled can only enlarge the support of the prior distribution, since if $p(\mathcal{R}, \mathcal{P}) > 0$, then the marginal distributions satisfy $p_R(\mathcal{R}) > 0, p_P(\mathcal{P}) > 0$. Thus, this assumption may lead to less effective estimation (as there are more MDP 'possibilities'), but should not prohibit the agent from estimating the true posterior with enough data.

We introduce **reward relabelling**, a simple solution to the MDP ambiguity problem. We propose to make the state distribution in the offline data approximately uniform across all MDPs, by replacing the rewards in a trajectory from some MDP $i$ in the data with rewards from another randomly chosen MDP $i' \neq i$. That is, for each $i \in 1, \ldots, N$, we add to the data $M$ trajectories $\hat{\tau}^{i,j}$, $j \in 1, \ldots, M$, where $\hat{\tau}^{i,j} = (s_0^{i,j}, a_0^{i,j}, \hat{r}_1^{i,j}, s_1^{i,j} \ldots, \hat{r}_H^{i,j}, s_H^{i,j})$, where the relabelled rewards $\hat{r}$ satisfy $\hat{r}_{t+1}^{i,j} = \mathcal{R}_{i'}(s_t^{i,j}, a_t^{i,j})$. Note that our relabelling effectively samples data from an MDP with transitions $\mathcal{P}_i$ and reward $\mathcal{R}_{i'}$, which has non-zero prior probability mass under the decoupled prior assumption. We only use the relabelled data for training the VariBAD VAE. While in principle it could also be used for training the off-policy RL algorithm, we did not find it useful in practice.

**MDP ambiguity due to transitions:** when ambiguity is due to different transitions, the problem becomes significantly more complicated, as we cannot simply 'relabel' trajectories in the data.[4] One solution is to add trajectories that run the policy for agent $i$ on task $i'$ (this is a generalization of the reward relabelling idea). This, of course, requires collecting more data, and is not suitable for the offline setting. It does, however, provide a guideline for effective data collection for OMRL. In this work, we focus on ambiguity due to rewards, and leave the question of MDP ambiguity due to transitions to future research.

## 4 RELATED WORK

Meta-learning considers training agents that quickly solve a new learning problem, by exploiting structure in the problem distribution (Thrun & Pratt, 1998; Hochreiter et al., 2001). In this work we focus on meta-RL – quickly learning to solve RL problems. Gradient based approaches to meta-RL seek policy parameters that can be updated to the current task with a few gradient steps (Finn et al., 2017; Grant et al., 2018; Rothfuss et al., 2018; Clavera et al., 2018). These are essentially online methods, and several studies investigated learning of structured exploration strategies in this setting (Gupta et al., 2018; Rothfuss et al., 2018; Stadie et al., 2018). Memory-based meta-RL, on the other hand, map the observed history in a task $h_{:t}$ to an action (Duan et al., 2016; Wang et al., 2016). These methods effectively treat the problem as a POMDP, and learn a memory based controller for it.

The connection between meta-learning and Bayesian methods, and between meta-RL and Bayesian RL in particular, has been investigated in a series of recent papers (Lee et al., 2018; Humplik et al., 2019; Ortega et al., 2019; Zintgraf et al., 2020), and our work closely follows these ideas. In particular, these works elucidate the difference between Thompson-sampling based strategies, such as in PEARL (Rakelly et al., 2019), and Bayes-optimal policies, such as in VariBAD (Zintgraf et al., 2020), and suggest to estimate the BAMDP belief using the latent state of deep generative models. *Our contribution is an extension of these ideas to the offline RL setting*, which to the best of our knowledge is novel. Technically, the VariBAD algorithm in Zintgraf et al. (2020) is limited to on-policy RL, and the off-policy method in Humplik et al. (2019) requires specific task descriptors during learning, while VariBAD, which our work is based on, does not.

---

[4]We remark that to our knowledge, previous work on online and offline meta-RL did not investigate such problems. For example, in the Walker environment of Zintgraf et al. (2020), the shape of the walker is varied, which manifests in almost every transition, and a successful agent must walk forward, thus many overlapping states are visited.

**Concurrently and independently with our work**, Li et al. (2020) proposed an offline meta-RL algorithm that combines BCQ (Fujimoto et al., 2018) with a task inference module. Interestingly, Li et al. (2020) also describe a problem similar to MDP ambiguity, and resolve it using a technique similar to reward relabelling. However, their approach does not take into account the task uncertainty, and cannot plan actions that actively explore to reduce this uncertainty – this is a form of Thompson sampling, where a task-conditional policy reacts to the task inference (see Figure 1 in Li et al. 2020). **Our work is the first to tackle offline meta-learning of Bayes-optimal exploration**. In addition, we demonstrate the first offline results on sparse reward tasks, which, compared to the dense reward tasks in Li et al. (2020), require a significantly more complicated solution than Thompson sampling (see experiments section). We achieve this by building on BRL theory, which both optimizes for Bayes-optimality and results in a much simpler algorithm. Recent work on meta Q-learning (Fakoor et al., 2019) also does not incorporate task uncertainty, and thus cannot be Bayes-optimal. The very recent work of Mitchell et al. (2020) considers a different offline meta-RL setting, where an offline dataset from the test environment is available.

Classical works on BRL are comprehensively surveyed in Ghavamzadeh et al. (2016). Our work, in comparison, allows training scalable deep BRL policies. Finally, there is growing interest in offline deep RL (Sarafian et al., 2018; Levine et al., 2020). Most recent work focus on how to avoid actions that were not sampled enough in the data. In our experiments, a state-of-the-art method of this flavor led to minor improvements, though future offline RL developments may possibly benefit OMRL too.

## 5   EXPERIMENTS

In our experiments, we aim to answer the following questions: (1) can we learn approximately Bayes-optimal policies in the offline setting? (2) what are the main practical challenges in OMRL? and (3) does our off-policy method improve meta-RL performance in the online setting as well?

Answering (1) is difficult because the Bayes-optimal policy is generally intractable, and because our results crucially depend on the available data. However, in deterministic domains with a single sparse reward, the optimal solution amounts to 'search all possible goal locations as efficiently as possible, and stay at goal once found; in subsequent episodes, move directly to goal.'. We therefore chose domains where this behavior can be identified qualitatively. Quantitatively, we compare our offline results with online methods based on Thompson sampling, which are not Bayes-optimal, and aim to show that the performance improvement due to being approximately Bayes-optimal gives an advantage even with the offline data restriction.

**Domains and evaluation metric:**   Our emphasis is on learning to explore efficiently. We thus focus on sparse reward problems where non-trivial exploration is required to identify the task: (1) A discrete $5 \times 5$ Gridworld (Zintgraf et al., 2020); (2) A continuous point robot where a sparse reward is located somewhere on a semi-circle (see Figure 1); (3) Ant-Semi-circle – a challenging modification of the popular Ant-Goal task (Fakoor et al., 2019) to a sparse reward setting similar to the semi-circle task above; (4) Half-Cheetah-Vel (Finn et al., 2017), a popular dense reward domain that serves as a comparison to the sparse domains (full details in Appendix C). Note that for all these problems, the tasks differ by their reward function. We also provide an experiment with varying transition functions in Appendix G.3. To evaluate performance, we measure average reward in the first 2 episodes on unseen tasks – this is where efficient exploration makes a critical difference.[5] In the supplementary, we report results for more evaluation episodes.

**Data collection and organization:**   For data collection, we used off-the-shelf DQN (Gridworld) and SAC (continuous domains) implementations. To study the effect of data diversity, we diversified the offline dataset by modifying the initial state distribution $P_{init}$ to either (1) uniform over a large region, (2) uniform over a restricted region, or (3) fixed to a single position. At meta-test time, only the single fixed position is used. The tasks are episodic, but we want the agent to maintain its belief between episodes, so that it can continually improve performance (see Figure 1). We follow Zintgraf et al. (2020), and aggregate $k$ consecutive episodes of length $H$ to a long trajectory of length $k \times H$, and we do not reset the hidden state in the VAE recurrent neural network after episode termination. For reward relabelling, we replace the first $k/2$ trajectories with trajectories from a randomly chosen

---

[5]For Gridworld, we measure average reward in the first 4 episodes.

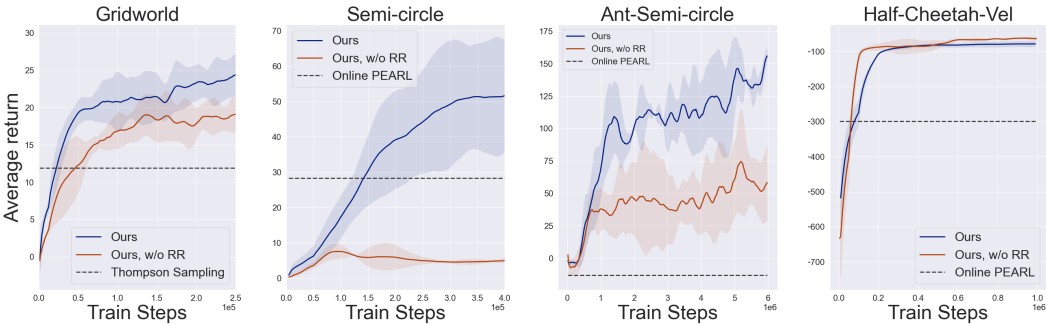

Figure 3: Offline performance on various domains. Blue: our method. Red: our method with reward relabelling ablated. Black: Thompson sampling baselines – calculated exactly in Gridworld, and using online PEARL for the other continuous domains.

MDP, and relabel their rewards. Technically, network architectures and hyperparameters were chosen similarly to Zintgraf et al. (2020), as detailed in the supplementary.

**Offline Results:** In Figure 3 we compare our offline algorithm with Thompson sampling based methods, and also with an ablation of the reward relabelling method. For Gridworld, the Thompson sampling method is computed exactly, while for the continuous environments, we use online PEARL (Rakelly et al., 2019) – a strong baseline that is not affected by our offline data limitation. In particular, this baseline is stronger than the offline meta-RL algorithm of Li et al. (2020). For these results the uniform initial state distribution was used to collect data. Note that **we significantly outperform Thompson sampling based methods, demonstrating our claim of learning non-trivial exploration from offline data**. Results on sparse reward domains signify the severity of MDP ambiguity – without reward relabelling performance drops significantly. In the Half-Cheetah-Vel environment, on the other hand, the rewards are dense and thus MDP ambiguity is not a problem.

In Figure 1 and in Figure 4, we visualize the trajectories of the trained agents in the Semi-circle and Ant-Semi-circle domains, respectively.[6] Qualitatively, an approximately Bayes-optimal behavior is evident: in the first episode, the agents search for the goal along the semi-circle, and in the second episode, the agents maximize reward by moving directly towards the already found goal. Similar behavior for Gridworld is reported in Appendix E. We emphasize that this behavior is very different from the training data, in which the agents learned to reach specific goals. In Figure 7 in the supplementary, we provide further insight into these results, by showing the belief update during the episode rollout: the belief starts as uniform on the semi-circle, and narrows in on the target as the agent explores the semi-circle. With reward relabelling ablated, however, we show that the belief does not update correctly.

---

[6]Video is provided: `https://youtu.be/6Swg55ZYOU4`

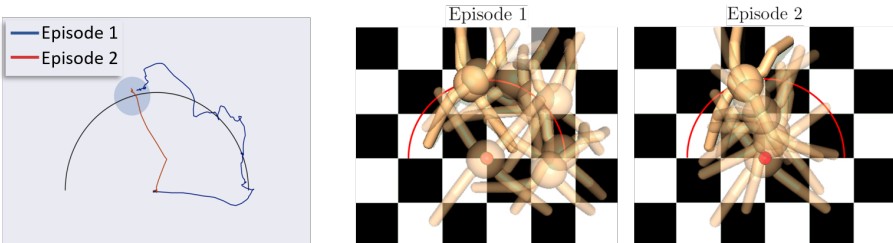

Figure 4: Ant-Semi-circle: trajectories from trained policy on a new goal. **Left:** Trajectory of the center of mass. **Right:** Visualization of the ant at different steps along the trajectories. Note that in the first episode, the ant searches for the goal, and in the second one it directly moves toward the goal it has previously found. This search behavior is different from the goal-reaching behaviors that dominate the training data.

**Data Quality Ablative Study:** To evaluate the dependency of our method on the offline data quality, we report results for the 3 different data collection strategies described above (see supplementary for more details). The results are summarized in Table 1. As expected, data diversity is instrumental to offline training. However, as we qualitatively show in Figure 10 in the supplementary, even on the low-diversity datasets, our agents learned non-trivial exploration strategies that search for the goal. This is especially remarkable for the fixed-distribution dataset, where it is unlikely that any training trajectory traveled along the semi-circle.

One may ask whether OMRL presents the same challenge as standard offline RL, and whether recent offline RL advances can mitigate the dependency on data diversity. To investigate this, we also compare our method with a variant that uses CQL (Kumar et al., 2020) – a state-of-the-art offline RL method – to train the critic network of the meta-RL agent. Interestingly, while CQL slightly improved results (Table 1), the data diversity is much more significant. Taken together with our results on MDP ambiguity, our investigation shows that OMRL presents different practical challenges than standard offline RL.

|  | Ours | w/ CQL |
|---|---|---|
| Uniform | $171.8 \pm 7.0$ | $176.0 \pm 10.2$ |
| Excluding s.c. | $102.8 \pm 32.7$ | $116.6 \pm 19.9$ |
| Fixed | $99.2 \pm 27.4$ | $112.4 \pm 31.3$ |

Table 1: Average return in Ant-Semi-circle for different initial state distributions during offline data collection: **Uniform** distribution, uniform distribution excluding states on the semi-circle (**Excluding s.c.**), and fixed initial position (**Fixed**).

**Online Setting:** Our method can also be applied to the online setting. In this case, it is simply a modification of VariBAD, where the policy gradient optimization is replaced with an off-policy RL algorithm. Since MDP ambiguity does not concern online meta-RL, we did not use reward relabelling in this setting. As shown in Figure 5, by exploiting the efficiency of off-policy RL, our method significantly improves sample-efficiency, without sacrificing final performance.

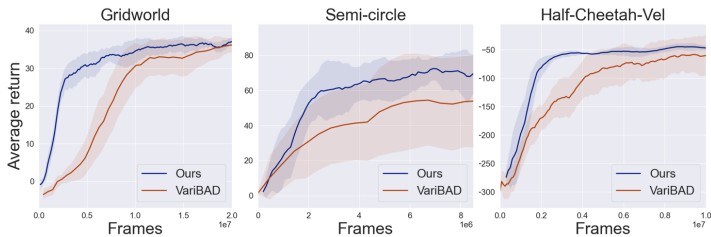

Figure 5: Online performance comparison. The off-policy optimization significantly improved VariBAD performance.

When comparing Figure 5 and Figure 3, the reader may notice that the online algorithm's final performance outperforms the final performance in the offline setting. We emphasize that this phenomenon largely depends on the quality of the offline data, and not on the algorithm itself.

## 6 CONCLUSION

We presented the first offline meta-RL algorithm that is approximately Bayes-optimal. Key to our approach is the connection between Bayesian RL and meta learning, which in principle allows to reduce the problem to standard offline RL. In practice, however, we showed that the MDP ambiguity problem prohibits learning, and proposed a simple solution based on mixing trajectories in the data and relabelling their rewards. Our results show that this solution is effective on several domains.

Offline learning is appealing for domains where data collection is costly, such as robotics and healthcare, and there is growing interest in applying deep RL to this setting (Levine et al., 2020). Key advances in this field will likely play a role in improving OMRL as well – an exciting direction for future research. How to handle the MDP ambiguity problem when it is caused by changes in the MDP transitions is another important research direction.

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

## A    PROPOSITION PROOF

For ease of reading, we copy Proposition 1 from Section 3.2:

**Proposition 1.** *Let $\tau = s_0, a_0, r_1, s_1 \ldots, r_H, s_H$ denote a random trajectory from a fixed history dependent policy $\pi$, generated according to the following process. First, MDP parameters $\mathcal{R}, \mathcal{P}$ are drawn from the prior $p(\mathcal{R}, \mathcal{P})$. Then, the state trajectory is generated according to $s_0 \sim P_{init}$, $a_t \sim \pi(\cdot|s_0, a_0, r_1, \ldots, s_t)$, $s_{t+1} \sim \mathcal{P}(\cdot|s_t, a_t)$ and $r_{t+1} \sim \mathcal{R}(s_t, a_t)$. Let $b_t$ denote the posterior belief at time t, $b_t = P(\mathcal{R}, \mathcal{P}|s_0, a_0, r_1, \ldots, s_t)$. Then*

$$P(s_{t+1}|s_0, a_0, r_1, \ldots, r_t, s_t, a_t) = \mathbb{E}_{\mathcal{R}, \mathcal{P} \sim b_t} \mathcal{P}(s_{t+1}|s_t, a_t), \text{ and,}$$
$$P(r_{t+1}|s_0, a_0, r_1, \ldots, s_t, a_t) = \mathbb{E}_{\mathcal{R}, \mathcal{P} \sim b_t} \mathcal{R}(r_{t+1}|s_t, a_t).$$

*Proof.* For the transitions, we have that,

$$P(s_{t+1}|s_0, a_0, r_0, \ldots, r_t, s_t, a_t) = \int P(s_{t+1}, \mathcal{R}, \mathcal{P}|s_0, a_0, r_0, \ldots, r_t, s_t, a_t) d\mathcal{R} d\mathcal{P}$$

$$= \int P(s_{t+1}|\mathcal{R}, \mathcal{P}, s_0, a_0, r_0, \ldots, r_t, s_t, a_t) P(\mathcal{R}, \mathcal{P}|s_0, a_0, r_0, \ldots, r_t, s_t, a_t) d\mathcal{R} d\mathcal{P}$$

$$= \mathbb{E}_{\mathcal{R}, \mathcal{P}} [P(s_{t+1}|\mathcal{R}, \mathcal{P}, s_0, a_0, r_0, \ldots, r_t, s_t, a_t)| s_0, a_0, r_0, \ldots, r_t, s_t, a_t]$$

$$= \mathbb{E}_{\mathcal{R}, \mathcal{P}} [\mathcal{P}(s_{t+1}|s_t, a_t)| s_0, a_0, r_0, \ldots, r_t, s_t, a_t]$$

$$= \mathbb{E}_{\mathcal{R}, \mathcal{P} \sim b_t} P(s_{t+1}|s_t, a_t).$$

The proof for the rewards proceeds similarly.    □

## B    VAE TRAINING OBJECTIVE

For completeness, we follow Zintgraf et al. (2020) and outline the full training objective of the VAE. Consider the approximate posterior $q_\phi(m|h_{:t})$ conditioned on the history up to time $t$. In this case, the ELBO can be derived as follows:

$$\log P(s_0, r_1, s_1 \ldots, s_H|a_0, \ldots, a_{H-1}) = \log \int P(s_0, r_1, s_1 \ldots, s_H, m|a_0, \ldots, a_{H-1}) dm$$

$$= \log \int P(s_0, r_1, s_1 \ldots, s_H, m|a_0, \ldots, a_{H-1}) \frac{q_\phi(m|h_{:t})}{q_\phi(m|h_{:t})} dm$$

$$= \log \mathbb{E}_{m \sim q_\phi(\cdot|h_{:t})} \left[ \frac{P(s_0, r_1, s_1 \ldots, s_H, m|a_0, \ldots, a_{H-1})}{q_\phi(m|h_{:t})} \right]$$

$$\geq \mathbb{E}_{m \sim q_\phi(\cdot|h_{:t})} [\log p_\theta(s_0, r_1, s_1 \ldots, s_H|m, a_0, \ldots, a_{H-1})$$
$$+ \log p_\theta(m) - \log q_\phi(m|h_{:t})]$$

$$= \mathbb{E}_{m \sim q_\phi(\cdot|h_{:t})} [\log p_\theta(s_0, r_1, s_1 \ldots, s_H|m, a_0, \ldots, a_{H-1})]$$
$$- D_{KL}(q_\phi(m|h_{:t})||p_\theta(m))$$

$$= ELBO_t(\theta, \phi).$$

The prior $p_\theta(m)$ is set to be the previous posterior $q_\phi(m|h_{:t-1})$, with initial prior chosen to be standard normal $p_\theta(m) = \mathcal{N}(0, I)$. The decoder $p_\theta(s_0, r_1, s_1 \ldots, s_H|m, a_0, \ldots, a_{H-1})$ factorizes to reward and next state models $p_\theta(s'|s, a, m)$ and $p_\theta(r|s, a, m)$, according to:

$$\log p_\theta(s_0, r_1, s_1 \ldots, s_H|m, a_0, \ldots, a_{H-1}) = \log p(s_0|m)$$
$$+ \sum_{t=0}^{H-1} [\log p_\theta(s_{t+1}|s_t, a_t, m) + \log p_\theta(r_{t+1}|s_t, a_t, m)].$$

As in Zintgraf et al. (2020), we trained only a reward decoder.

The overall training objective of the VAE is to maximize the sum of ELBO terms for different time steps,

$$\max_{\theta, \phi} \sum_{t=0}^{H} ELBO_t(\theta, \phi). \tag{4}$$

## C    ENVIRONMENTS DESCRIPTION

In this section we describe the details of the domains we experimented with.

**Gridworld:**    A $5 \times 5$ gridworld environment as in Zintgraf et al. (2020). The task distribution is defined by the location of a goal, which is unobserved and can be anywhere but around the starting state at the bottom-left cell. For each task, the agent receives a reward of $-0.1$ on non-goal cells and $+1$ at the goal, i.e.,

$$r_t = \begin{cases} 1, & s_t = g \\ -0.1, & \text{else,} \end{cases}$$

where $s_t$ is the current cell and $g$ is the goal cell.
Similarly to Zintgraf et al. (2020), the horizon for this domain is set to 15 and we aggregate $k = 4$ consecutive episodes to form a trajectory of length 60.

**Semi-circle:**    A continuous 2D environment as in Figure 1, where the agent must navigate to an unknown goal, randomly chosen on a semi-circle of radius 1 (Rakelly et al., 2019). For each task, the agent receives a reward of $+1$ if it is within a small radius $r = 0.2$ of the goal, and 0 otherwise,

$$r_t = \begin{cases} 1, & \|x_t - x_{\text{goal}}\|_2 \leq r \\ 0, & \text{else,} \end{cases}$$

where $x_t$ is the current 2D location. Action space is 2-dimensional and bounded: $[-0.1, 0.1]^2$.
We set the horizon to 60 and aggregate $k = 2$ consecutive episodes to form a trajectory of length 120.

**MuJoCo:**    For both MuJoCo domains, the horizon is set to $H = 200$ and we consider $k = 2$ episodes.

1. **Half-Cheetah-Vel:** In this environment, a half-cheetah agent must run at a fixed target velocity. Following recent works in meta-RL (Finn et al., 2017; Rakelly et al., 2019; Zintgraf et al., 2020), we consider velocities drawn uniformly between 0.0 and 3.0. The reward in this environment is given by

$$r_t = -|v_t - v_{\text{goal}}| - 0.05 \cdot \|a_t\|_2^2$$

   where $v_t$ is the current velocity, and $a_t$ is the current action.

2. **Ant-Semi-circle:** In this environment, an ant needs to navigate to an unknown goal, randomly chosen on a semi-circle, similarly to the Semi-circle task above.

   When collecting data for this domain, we found that the standard SAC algorithm (Haarnoja et al., 2018) was not able to solve the task effectively due to the sparse reward (which is described later), and did not produce trajectories that reached the goal. We thus modified the reward **only during data collection** to be dense, and inversely proportional to the distance from the goal,

$$r_t^{\text{dense}} = -\|x_t - x_{\text{goal}}\|_1 - 0.1 \cdot \|a_t\|_2^2$$

   where $x_t$ is the current 2D location and $a_t$ is the current action. After collecting the data trajectories, we replaced all the dense rewards in the data with the sparse rewards that are given by

$$r_t^{\text{sparse}} = -0.1 \cdot \|a_t\|_2^2 + \begin{cases} 1, & \|x_t - x_{\text{goal}}\|_2 \leq 0.2 \\ 0, & \text{else.} \end{cases}$$

   We note that Rakelly et al. (2019) use a similar approach to cope with sparse rewards in the online setting.

## D    EXPERIMENTAL DETAILS

In this section we outline our training process and hyperparameters.

For the discrete Gridworld domain we used DQN (Mnih et al., 2015) with soft target network updates, as proposed by Lillicrap et al. (2015), which has shown to improve the stability of

learning. For the rest of the continuous domains, we used SAC (Haarnoja et al., 2018) with the architectures of the actor and critic chosen similarly, and with a fixed entropy coefficient. For both DQN and SAC, we set the soft target update parameter to 0.005.

In all our experiments we average the performance over 3 random seeds and present the mean and standard deviation.

### D.1 OFFLINE SETTING

Our offline training procedure is comprised of 3 separate training steps. First is the training of the data collection RL agents. Each agent is trained on a different task from the task distribution.

For the Gridworld domain, we train 21 agents. We note that this covers the entire task distribution, as goals can be anywhere but around the starting state at the bottom-left cell. For the Semi-circle and Ant-Semi-circle domains, we train 80 data collection agents, and for the Half-Cheetah-Vel environment we used 100 agents.

For all tasks, we used a similar architecture of 2 fully-connected (FC) hidden layers of size that depends on the domain with ReLU activations, and set the batch size to 256. The rest of the hyperparameters used for training the data collection RL agents are summarized in the following table:

| | Gridworld (DQN) | Semi-circle (SAC) | MuJoCo (SAC) |
|---|---|---|---|
| Hidden layers size | 16 | 32 | 128 |
| Num. iterations | 200 | 300 | 1000 |
| RL updates per iter. | 500 | 500 | 2000 |
| Exploration/entropy coeff. | $\epsilon$-greedy, linear annealing from 1 to 0.1 over 100 iterations | 0.01 | 0.2 |
| Collected episodes per iter. | 5 | 2 | 2 |
| Learning rate/s | $3 \cdot 10^{-4}$ | $3 \cdot 10^{-4}$ | $3 \cdot 10^{-4}$ |
| Discount factor ($\gamma$) | 0.99 | 0.9 | 0.99 |

The second training step is the VAE training after applying reward relabelling to the collected data.

The VAE consists of a recurrent encoder, which at time step $t$ takes as input the tuple $(a_t, r_{t+1}, s_{t+1})$. The state and reward are passed each through a different fully-connected (FC) layer. The state FC layer is of size 32 and the reward FC layer is of size 8 for the Gridworld and 16 for the rest of the domains, all with ReLU activations. For the MuJoCo environments, we also pass the action through a FC layer of size 16 with ReLU. Then, the state and reward layers' outputs are concatenated along with the action (or with the output of the action layer in the case of MuJoCo) and passed to a GRU of size 64/128 (Gridworld/other domains). The GRU outputs the Gaussian parameters $\mu(h_{:t}), \Sigma(h_{:t})$ of the latent vector $m$, whose dimensionality is 5 in all our experiments.

The VAE reward decoder takes as input a latent sample $m \sim \mathcal{N}(\mu(h_{:t}), \Sigma(h_{:t}))$ and the states along the trajectory $s_1, \ldots, s_H$, each state at a time, and outputs (for every timestep $t = 1, \ldots, H$) the entire reconstructed/predicted rewards $r_1, \ldots, r_H$ along the trajectory. In the MuJoCo domains the reward decoder also takes as input the actions along the trajectory $a_1, \ldots, a_H$ and the previous states $s_0, \ldots, s_{H-1}$, as the reward $r_t$ in these environments generally depends on $s_{t-1}, a_t, s_t$. The reward decoder is comprised of 2 FC layers, each of size 32.

The VAE is trained to optimize Equation 4, but similarly to Zintgraf et al. (2020), we weight the KL term in each of the ELBO terms with some parameter $\beta$, which is not necessarily 1. In our experiments we used $\beta = 0.05$.

After the VAE is trained, we apply state relabelling to the data collected by the RL agents, to create a large offline dataset that effectively comes from the BAMDP. Then, we train an off-policy RL algorithm, which is our meta-RL agent, using the offline data.

For the offline meta-RL agents training, we used similar hyperparameters to those used for the data collection RL agents training. We only enlarge the size of the hidden layers in our models from 16, 32 and 128 to 64, 128 and 256 for the Gridworld, Semi-circle and MuJoCo domains, respectively. In

every iteration we perform 1000 parameter updates for all environments expect the Ant-Semi-circle, in which case we perform 2000 updates per iteration.

## D.2 ONLINE SETTING

In the online setting we didn't apply reward relabelling to the data, since, as we explained, MDP ambiguity doesn't concern online meta-RL. The hyperparameters used in the online setting are as follows:

| | Gridworld (DQN) | Semi-circle (SAC) | Cheetah-Vel (SAC) |
|---|---|---|---|
| **RL parameters** | | | |
| Architecture/s | 2 FC layers of size 100. | 2 FC layers of size 128. | 3 FC layers of size 128. |
| Num. updates per iter. | 250 | 1000 | 2000 |
| Exploration/entropy coeff. | $\epsilon$-greedy, linear annealing from 1 to 0.1 over 1000 iterations. | 0.01 | 0.2 |
| Collected episodes per iter. | 25 | 25 | 25 |
| Learning rate/s | $7 \cdot 10^{-5}$ | $7 \cdot 10^{-5}$ | $3 \cdot 10^{-4}$ |
| Discount factor ($\gamma$) | 0.99 | 0.9 | 0.99 |
| **VAE parameters** | | | |
| Encoder architecture | state/reward FC layer of size 32/8. GRU of size 64. | state/reward FC layer of size 32/8. GRU of size 128. | state/action/reward FC layer of size 32/16/16. GRU of size 128. |
| Reward decoder architecture | 2 FC layers of size 32. | 2 FC layers of sizes 64 and 32. | 2 FC layers of sizes 64 and 32. |
| Num. updates per iter. | 20 | 25 | 20 |
| Learning rate | $3 \cdot 10^{-4}$ | $10^{-3}$ | $3 \cdot 10^{-4}$ |
| Weight of KL term ($\beta$) | 1.0 | 0.1 | 1.0 |

## E LEARNED BELIEF VISUALIZATION

In this section we visualize the learned belief states, in order to get more insight into the decision making process of the agent during interaction.

In Figure 6, we visualize the interaction of a trained agent with the Gridworld environment, exactly as visualized in Figure 3 at Zintgraf et al. (2020). The agent reduces its uncertainty by effectively searching the goal. After the goal is found, the agent stops and in subsequent episodes it directly moves toward it.

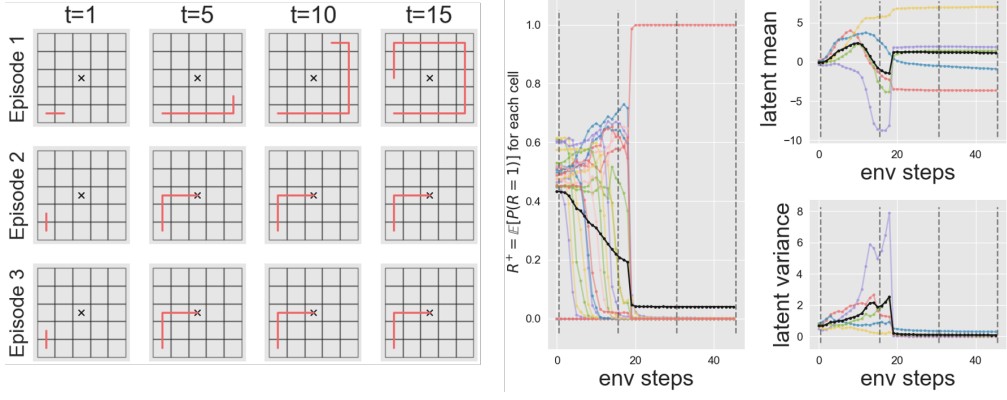

Figure 6: Interaction of trained agent with Gridworld. For more details, see Zintgraf et al. (2020).

In Figure 7, we plot the reward belief (obtained from the VAE decoder) at different steps during the agent's interaction in the Semi-circle domain. Note how the belief starts as uniform over the

semi-circle, and narrows in on the target as more evidence is collected. Also note that without reward relabelling, the agent fails to find the goal. In this instance of the MDP ambiguity problem, the training data for the meta-RL agent consists of trajectories that mostly reach the goal, and as a result, the agent believes that the reward is located at the first point it reaches on the semi-circle.

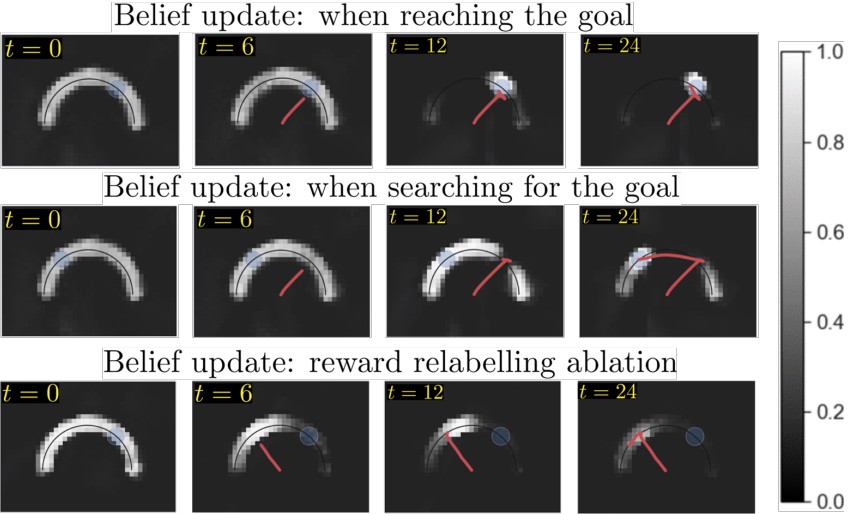

Figure 7: Semi-circle belief visualization. The plots show the reward belief over the 2-dimensional state space (obtained from the VAE) at different stages of interacting with the system. The red line marks the agent trajectory, and the light blue circle marks the true reward location. **Top:** Once the agent finds the true goal, it reduces the belief over other possible goals from the task distribution. **Middle:** As long as the agent doesn't find the goal, it explores efficiently, reducing the uncertainty until the goal is found. **Bottom:** Without reward relabelling, the agent doesn't learn to differentiate between different MDPs, and therefore fails to identify the goal.

## F    DATA QUALITY ABLATION

In our data quality ablative study, we consider the Ant-Semi-circle domains, for which we modify the initial state distribution during the data collection phase. The initial state distributions we consider are visualized in Figure 8: Uniform distribution, uniform excluding states on the semi-circle, and fixed initial position.

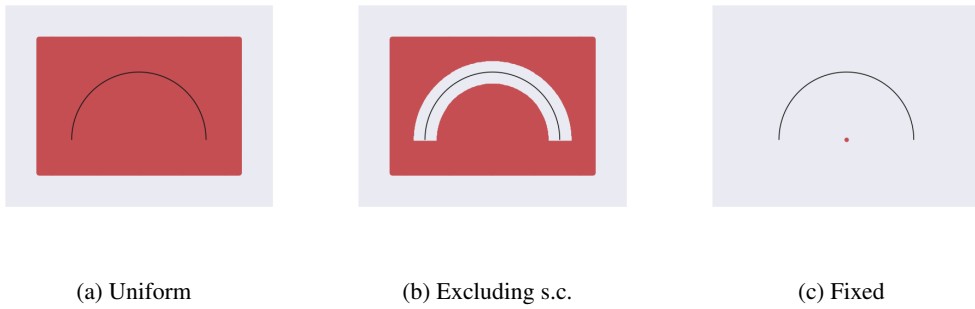

(a) Uniform                    (b) Excluding s.c.                    (c) Fixed

Figure 8: Initial state distributions. Red locations indicate non-zero sampling probability.

Figure 9 shows the learning curves for the results presented in Table 1. For completeness, we add the learning curve for the uniform distribution which is also presented in Figure 3.

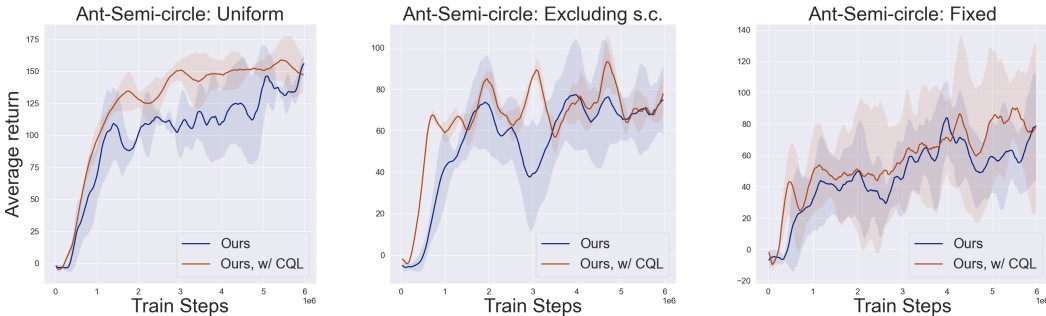

Figure 9: Learning curves for the results presented in Table 1. In blue is our method and in red is our method, with critic network trained according to the CQL objective (Kumar et al., 2020). **Left:** Uniform initial state distribution. **Middle:** Uniform distribution, excluding states over the semi-circle. **Right:** Initial state is fixed.

We also visualize trajectories of trained agents for the 3 different cases as well as for PEARL (Rakelly et al., 2019), in Figure 10. Note that even for the fixed-distribution dataset, our agent learns to search for the goal.

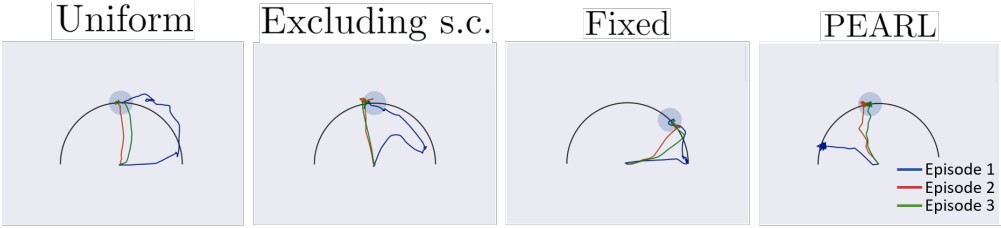

Figure 10: Ant-Semi-circle: trajectories of trained agents for different offline datasets and for PEARL.

## G  ADDITIONAL RESULTS

### G.1  PERFORMANCE VS. ADAPTATION EPISODES

In this part, we present the average reward per-episode as a function of the number of adaptation episodes at the environment. Figure 11 shows the performance for the Ant-Semi-circle and Half-Cheetah-Vel domains. Note that within the first few episodes, PEARL does not collect high rewards due to the Thompson sampling-based nature of the algorithm. Our method, on the other hand, efficiently explores new tasks and is able to collect rewards within the first episodes of interaction.

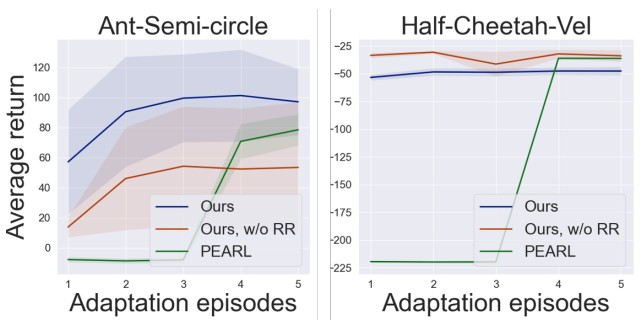

Figure 11: Adaptation performance. Our method outperforms PEARL, collecting high rewards within the first adaptation episodes.

### G.2  PEARL LEARNING CURVES

We present the training curves of PEARL in Figure 12. Note that since PEARL is an online algorithm, the $x$-axis represents the number of environment interactions.

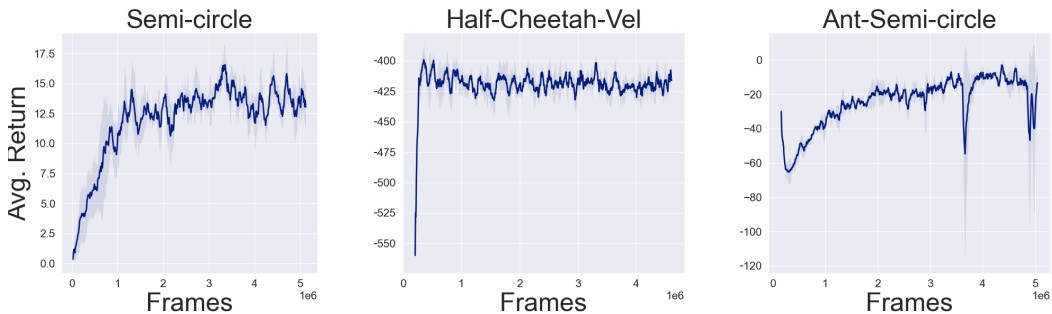

Figure 12: Learning curves for online PEARL training. Performance is measured by average reward in the first 2 episodes on unseen tasks from the task distribution.

### G.3 EXPERIMENT ON MDPs WITH VARYING TRANSITIONS

In the following section we describe experiments on a domain in which MDPs differ from each other by their transition function, when MDP ambiguity is not a problem. Our goal is to show that our method can also be applied to MDPs with varying transitions.

We consider a continuous 2-dimensional environment we term Point-Robot-Wind. In Point-Robot-Wind, the agent must navigate to a fixed (unknown) goal within a distance of $D = 1$ from the initial state. The goal position is the same for all tasks. The agent receives a reward of $+1$ if it is within a radius $r = 0.2$ of the goal, and 0 otherwise. For each task in this domain, the agent is experiencing a different 'wind', which results in a shift in the transitions, such that when taking action $a_t \in [-0.1, 0.1]^2$ from state $s_t$ in MDP $\mathcal{M}$, the agent transitions to a new state $s_{t+1}$, which is given by

$$s_{t+1} = s_t + a_t + w_{\mathcal{M}},$$

where $w_{\mathcal{M}}$ is a task-specific wind, which is randomly drawn for each task from the uniform distribution on $[-0.05, 0.05]^2$. To navigate correctly to the goal and stay there, the agent must take actions that cancel the wind effect.

We set the horizon to 25 and evaluate the performance in terms of average return within the **first** episode of interaction on test tasks, sampled from the task distribution. We do not apply reward-relabelling, as the MDPs differ in their transition functions rather than rewards. In Figure 13(a) we compare our method with online PEARL. Note that our learned policy performs much better than the online PEARL baseline.

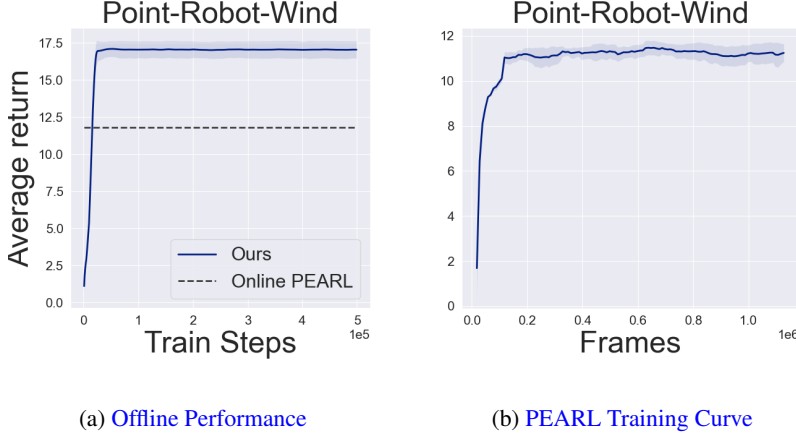

(a) Offline Performance        (b) PEARL Training Curve

Figure 13: Evaluation on Point-Robot-Wind – domain with varying transition function. In (a) is offline performance of our method, compared to the best performance of online PEARL (similar to Figure 3), and in (b) is the learning curve for online PEARL training (similar to Figure 12).

In Figure 14 we visualize trajectories of the trained agent on different test tasks. As can be seen, after several steps in the environment, our agent learns to adapt to the varying wind, and travels to the goal in a straight line. PEARL, on the other hand, only adapts after the first episode, and therefore obtains worse results. We believe it is possible to improve PEARL to update its posterior after every step, and in this case the improved PEARL will obtain similar performance as our method in Point-Robot-Wind. However, this will not work in the sparse reward domains described in the main text, where the Bayes adaptive exploration has an inherent advantage over Thompson sampling. We emphasize that in Point-Robot-Wind, MDP ambiguity is not a concern, since the data from all agents is largely centered on the line between the agent's initial position and the goal. Thus, the effect of the wind on these states can uniquely be identified in each domain. These results confirm that when MDP ambiguity is not a concern, our method learns an effective policy in domains with varying transitions.

Figure 14: Point-Robot-Wind: trajectories of trained agent on different test tasks from the task distribution.

