# OpenReview forum: "Offline Meta Learning of Exploration"
_ICLR.cc/2021/Conference — Reject_

### Official Review · AnonReviewer3 · 2020-10-27
**Good submission, but central claim supported by evidence**

**Rating:** 7
**Confidence:** 4

**Review:**

Summary after discussion period:
----------------------------------------------
The authors have done a good job in toning down their claims to match what the evidence supports. After reading the other review's comments and the updated version of the paper, I feel both my comments and most of the reivewer's comments have been addressed, allowing me to recommend this paper for acceptance.

Summary
-------------

In this paper, the authors tackle the problem of offline meta RL, where one observes trajectories from environments drawn from a distribution of environments, then using these past observations finds a policy which will work well on new environments drawn from the same distribution, optimizing both in the face of uncertainty about the environment and about the random transitions. The authors propose a method based on VariBAD which achieves good experimental results.

Pros:
-------

I find the paper to be novel, relevant and well written, a good contribution to the scientific community. I liked it’s introduction to the idea of offline Meta-RL, and thought the experimental results where an agent learns to walk the semi-circle were impressive. Further, I greatly enjoyed the discussion of MDP ambiguity, it’s a genuine problem that I’d never seen before and I think addressing exactly this issue will be important in advancing offline meta-RL.

Major Concern:
--------------------

My major concern is that the evidence does not fully support the author’s central claim: “quickly maximize reward in a new, unseen task from the same distribution” where this distribution is described as varying over both the reward and transition functions. Although I do believe that the author’s method addresses varying reward structures, they have provided no evidence that the method addresses varying transition dynamics. In their experiments, they sample different MDPs, but each has the same transition dynamics, only the reward dynamics vary.

Please do not misunderstand me, I believe that science is incremental, small steps are valuable, and having a method which addresses only the smaller problem of fixed-transition dynamics with sampled reward dynamics is already an interesting and valuable contribution. Yet I don’t feel anyone is served by a paper which claims it solves the larger problem of offline Meta-RL, but doesn’t provide evidence for this larger claim.

Therefore I would request the authors either provide evidence for the larger claim, or (it’s probably the easier route) simply make a smaller claim that is supported by the evidence, and save the real offline Meta-RL problem for a future submission. This would be that one is interested in the setting where MDPs with varying reward structure are sampled.

I believe that the problem of MDP ambiguity can’t be so easily solved for ambiguous state transitions. In section 3.3, the authors nicely address the issue of MDP ambiguity, (it was a very insightful section, a valuable contribution) and that it’s impossible to know if two trajectories come from different MDPs with different rewards, or one MDP with rewards at both locations. It’s an inherent counterfactual problem, one doesn’t know what would have happened if one had done something else.

The authors fill in the gaps in this counterfactual knowledge by assuming knowledge of the true reward structure from each past MDP. In this way, for all past trajectories the authors can say what would have happened if the same actions had been taken in a setting with a different rewards structure.

Yet the problem of MDP ambiguity isn’t restricted to unknown reward structures. Assume a meta MDP setting where each MDP is a gridworld with a key and a door and the agent collects reward after getting the key then going to the door, thus the state is the agent’s position and a binary variable indicating whether the agent has collected the key. For different MDPs in the distribution, the key is placed in different places. In this way, the reward structure is the same for different MDPs in the distribution, but the transition structure (where the agent collects the key) changes. I don’t see how the author’s proposed MDP ambiguity fix would address ambiguity in the transition dynamics, or even provide a roadmap for addressing this issue.

Indeed, it is telling that the experiments only deal with Meta-RL where the reward distribution changes, and never address changing transition dynamics.

Therefore I would ask the authors to clarify the scope of their contribution, and perhaps address what challenges stand in the way of offline Meta-RL where the state transitions vary too. After that, I'd be happy to update my review.

Minor Concerns:
----------------------
In Prop. 1, why do we have $E_{\mathcal R, \mathcal P\sim b_t}$, it seems one must only sample $\mathcal P$ in the first line, and $\mathcal R$ in the second.

“Recall that the VariBAD VAE encoder….” perhaps drop the word ‘encoder’, encoder is already in the ‘E’ of VAE.

In section 3.3, when describing figure 2 (also in the caption of figure two) you say the reward is in the yellow circle, but the circle looks reddish on my PDF viewer.

---

> ### Author Response · Authors · 2020-11-18
> **Reply to reviewer 3**
>
> We completely agree with the reviewer's evaluation of our work -- both the merits and the limitations. Regarding MDP ambiguity - please see our general comment above. Our method should successfully handle MDPs with different transitions, but only if the identifiable states are dense enough. Indeed, when ambiguity is due to transitions, our method is not applicable, and we do not see how this problem can be solved. We will clarify this issue in the text.
>
> We are running experiments with a 'dense' uncertainty in transitions. If the results will not be ready before the rebuttal period ends, we will remove our claims of handling varying transition functions.

---

### Official Review · AnonReviewer2 · 2020-10-28
**Offline Meta-RL using a variational approach coupled with state and reward relabeling. Discusses an important setting, and some new algorithmic ideas therein, but whether these ideas will work in practice is inconclusive.**

**Rating:** 5
**Confidence:** 3

**Review:**

The paper studies the problem of offline Meta-Reinforcement Learning (RL). In this problem, N separate RL environments are considered which are drawn from a specific underlying distribution. For each such environment, M trajectories each of horizon H are provided beforehand. The task is to train an RL agent that performs well in expectation on a new RL environment drawn from the same distribution.  The authors adapt the online VariBAD algorithm (Zintgraf et al., 2020) by the use of a techniques called state relabeling and reward relabeling.

The VariBAD algorithm formulates online Meta-RL as a Bayesian Adaptive MDP (BAMDP) and solves it approximately using a variational approach. The authors, due to lack of online data, use partial offline trajectories to create belief over RL environment. Next, this belief is used to map offline Meta-RL to the BAMDP setting -- a.k.a. state relabelling. Once mapped to BAMDP off-the-shelf offline RL techniques are used to solve the problem approximately. The authors further introduce reward relabelling that adds new trajectories by mixing rewards among the RL environments conditioned on the state and action. This is claimed to be useful if the reward distribution is independent of the transition matrix distribution of the RL.

The authors provide multiple experiments with sparse rewards to show the VariBAD with relabelling works in some established benchmark RL tasks (albeit in the new offline Meta-RL setting).

Pros:
* The offline Meta-RL is a very relevant problem with a lot of applications in the coming years. Designing a methodology to solve this problem using off-the-shelf techniques is a novel pursuit, which is carried out in the paper.

* The idea of state and reward relabeling is novel in the meta-RL setup, as per my understanding (as an outsider to the Meta-RL community but with a background in RL).

Cons:
1. Methodology:

* The paper does not explore properly the (in)-famous "deadly triad" which is known to make offline RL difficult. Given the state-space of BAMDP captures beliefs over (possibly) complicated RL environments, the effect of  "deadly triad" is conceivably even more prominent in the current formulation.

* I am not sure why the reward relabeling is able to solve the MDP (a.k.a. aliasing in the offline RL community). In particular, "Note that our relabelling effectively samples data from an MDP with transitions Pi and reward Ri, which has non-zero prior probability mass under the decoupled prior assumption" -- the above statement is very vague given finite trajectories. The MDP aliasing (a part of the deadly triad) is closely related to this issue.

* The state relabelling is heavily reliant on the existing VariBAD algorithm, and off-the-shelf RL techniques. Therefore, the algorithmic contribution seems somewhat lacking in my opinion.

2. Experiments:
* The provided experiments lack any conclusive evidence of the effectiveness of reward-labeling. Why in the Half-Cheetah-Vel experiment reward relabeling performs worse -- is the reward independence assumption not true here?  Will reward relabeling always perform worse?

* The effectiveness of the Thompson sampling baseline used is not clear to me. Does no other simple baseline exist? I am alright with not comparing it with the concurrent works. However, such a comparison would have been much more convincing.

---

> ### Author Response · Authors · 2020-11-18
> **Reply to reviewer 2**
>
> **Methodology**: \
> Deadly triad: the deadly triad concerns theoretical convergence issues when bootstrapping and function approximation are used in off-policy RL. In recent years, many successful applications of off-policy RL with deep neural networks have been reported. Thus, it seems that with some implementation tricks (e.g., experience replay, target networks), it is not a practical limitation. The off-policy algorithms we use (DQN, SAC) are well documented to solve challenging RL problems with deep neural networks and high dimensional state spaces.
> The belief in our case is represented by the means and covariances from the VAE, which can be seen as simply additional continuous state variables. As our results indicate, the RL algorithms we used were capable of handling this scenario.
>
> Aliasing and MDP ambiguity: state aliasing occurs when two states have similar representation under the RL function approximator and, as a result, when we update the Q-function in one of them, we also change the Q-function of the other. The MDP ambiguity problem, on the other hand, affects the training of the VAE, and *is not a concern for the off-policy RL solver*. The goal of the VAE is to infer an MDP from experience (represented by Gaussian approximate posterior). When MDP ambiguity occurs, inferring the correct MDP becomes impossible since, based on offline data, we cannot tell from which MDP the data came from (no overlapping `identifying states' -- see general comment above). Therefore, this problem prevents us from obtaining a good estimate of the belief (which is part of the hyper-state we form).
>
> Reward relabelling: When we relabel the rewards of a state-action trajectory collected from MDP i, by querying the reward function of MDP i', we effectively add a state-action-reward trajectory that is collected from an MDP with transitions P_i (s^(i)-a^(i)-s^'(i)) and rewards R_i' (s^(i)-a^(i)-r^(i')). Such MDP (with parameters P_i, R_i') has non-zero probability under the prior (it lies within the prior support).
>
> Algorithmic contribution: as we acknowledge in the paper, the state-relabelling is indeed similar to that used in VariBAD, and we could have just run VariBAD blindly with off-policy RL and gotten the same results. However, understanding why the algorithm works is important. Our contribution is providing the theory for VariBAD in the off-policy setting, which is conceptually different from standard VariBAD due to the fact that off-policy methods are trained using tuples of (s,a,r,s') rather than trajectories. See Section 3.2 for more details.
>
>
> **Experiments**: \
> Half-cheetah domain: please see our general comment about MDP ambiguity and sparse rewards, and the explanation why in dense rewards domains (as with Cheetah), reward relabelling is not needed. The performance with reward relabelling is only slightly worse than without it, and we believe the differences are due to noise (random seed) and are not systematic. We emphasize that both with and without reward relabelling, our method outperforms the PEARL baseline.
>
> Baselines: to the best of our knowledge, there are no studies that meta learn exploration strategies in the offline setting that are not based on Thompson sampling (TS), *including concurrent works*. Thus, by comparing with a state-of-the-art TS algorithm that is trained *online* (or computed exactly in the discrete setting), we effectively compare with an *upper bound to the performance of all other works in the offline setting to date*.

---

> > ### Comment · AnonReviewer2 · 2020-11-18
> > **The paper seems to be effective for sparse reward problems, where reward for each (s,a) has same distribution in all MDPs**
> >
> > Thanks for the response. I will maintain my score due to the following reasons.
> >
> > From the response, it is apparent that the proposed techniques do not resolve the MDP aliasing issue as it mainly handled by the training of the VAE. This limits the importance of the paper to some extent.
> >
> > In the general comment, I do not see any justification behind "Our approach is not limited to problems with reward uncertainty, and we expect it to work on MDPs with varying transitions." Again, this highlights the disconnect between the claims and the explanations provided in the paper as mentioned by Reviewer3.
> >
> > Further, I believe, the reward relabeling works because it is assumed that reward r ~ R(s,a) irrespective of the underlying MDP (in Proposition 1).  Note, here I have assumed that s is not the augmented state, as the authors use s^+ to denote the augmented state.
> > If we need to make the reward for a state s dependent on the underlying MDP, we need to assume that r ~ R(s^+,a).  *Please correct me* if my interpretation is wrong here.  But if this interpretation is correct it may limit the contribution further.

---

> > > ### Author Response · Authors · 2020-11-19
> > > **Reply**
> > >
> > > From the reviewer’s comments it seems that main points in our contribution were missed.
> > >
> > > Reward independent of MDP - this is completely incorrect, and contradicts our fundamental problem setting! Even from the illustration in Figure 1 it is clear that different MDPs have different rewards, and this is the basic idea in all the discussion above on MDP ambiguity. Further, Proposition 1 is not related to reward relabelling.
> > > Perhaps we did not understand the reviewer’s question correctly?
> > >
> > > Our approach is not limited to problems with reward uncertainty: we ask the reviewer to examine again our explanation in the general comment. There is a distinction between problems with MDP ambiguity and without MDP ambiguity. In problems without ambiguity, the fact that there are varying transitions should not be a problem for our method, for the same reason that Half-Cheetah worked without reward relabelling. Reviewer 3’s comment concerns problems where the set of identifying transitions is sparse and causes MDP ambiguity.
> > >
> > > The proposed techniques do not resolve the MDP aliasing issue - we are not certain what the reviewer means by aliasing. The MDP ambiguity problem, which is a special problem to offline meta-RL, and to our knowledge was first discussed in this work, is addressed by our solution for the case of varying rewards. It is not related to the aliasing problem in standard RL due to function approximation, as we discussed in our previous comment to the reviewer.
> > >
> > > We would like to further emphasize our contribution: we are the first to consider the offline setting of meta learning exploration, and our results, even if limited to certain scenarios, allow to solve problems that were not possible before. Our work also makes clear what are the main challenges in this problem, which may be solved in future work. Any such future investigation, we believe, will make use of our ideas, experiments, and datasets (at least as a baseline).

---

> > > > ### Comment · AnonReviewer2 · 2020-11-24
> > > > **Reply: Justify why the assumptions made are reasonable**
> > > >
> > > > I agree that the reward is MDP dependent. In proposition 1, I was confused by the notation E_{R,P∼b_t}, which I understood as only P being distributed as b_t. From the response, I gather it implies both (R,P) is distributed as b_t.
> > > >
> > > > I missed the reason why reward relabeling works, which is already there in the paper. It works because of the assumption "(1) The transition and reward uncertainties are decoupled". This ensures that the relabeled reward is also a valid MDP.  The authors should make the assumptions in the paper more noticeable.  In this regard, the claim "these assumptions are largely satisfied in most meta-RL studies to date" must be justified by proper citations.
> > > >
> > > > The aliasing issue that I mention is related to function approximation in RL. As VariBAD approximates the sufficient statistics using a Gaussian function, there is an element of function approximation used to represent the belief (a part of the state).

---

> > > > > ### Author Response · Authors · 2020-11-24
> > > > > **Reply**
> > > > >
> > > > > Thank you for your response. We will emphasize the assumption and duly add citations to our claim regarding decoupled uncertainty.
> > > > >
> > > > > Aliasing: you are correct, the VAE only provides an approximation of the belief. The approximation is due to both function approximation and the fact that it is trained using a variational approximation (lower bound on the likelihood). Calculating the true belief is only possible for very specific systems (e.g., linear Gaussian dynamics, where the posterior is known to be Gaussian).
> > > > > We provided an analysis of the learned posterior for the grid world and semi-circle domain — *please check out Section E* in the supplementary. We provided plots of the posterior, and an ablation study for the case of no reward relabelling. These plots show that our learned belief is rather accurate when reward relabelling is applied, but is inaccurate without it. These results are qualitative, as obtaining quantitative measurements of the belief in the semi-circle domain is intractable. Still, we believe that they provide important insight into what the VAE learns. In particular, they show that, at least in this case, a qualitatively correct posterior is obtained even though we use approximations.

---

### Official Review · AnonReviewer1 · 2020-11-02
**Review for OFFLINE META LEARNING OF EXPLORATION**

**Rating:** 6
**Confidence:** 3

**Review:**

Summary:

The paper proposes an extension of VariBAD to the offline setting. The main difficulty of such an application in an offline setting is what the paper termed "MDP ambiguity" while proposing reward relabeling as a solution.

#########################################################

 Pros:

1. The paper is clear and well-written.
2. It proposes a simple but effective solution to adapt VariBAD to the offline setting for a type of environments  where transition probabilities and reward probabilities are independent.
3. The paper demonstrates very interesting learned exploration behavior and shows its superior empirical performance compared to baselines (VariBAD and PEARL).

#########################################################

Cons:

1. The test environments are not extensive: all 4 test environments used in the paper are quite simple and 2 out of 4 are even similar. What about other openai gym control tasks (or Atari games)? Given the strong independence assumption between transition and reward probabilities, it would be nice to demonstrate how resilient this method is on environments that violate this assumption.

2. There are few baselines being compared against in this paper so it's unclear given some offline data from an unknown environment this method is the sota. For example, one potential area of comparison is model-based RL agents where the prediction models (or even transition models if the environment is simple enough) are trained with supervised learning using offline data.

#########################################################

 Questions during rebuttal period:

1. Why is the top right slot of Table 1 blank?

---

> ### Author Response · Authors · 2020-11-18
> **Reply to reviewer 1**
>
> To the best of our knowledge, the Ant-Semi-circle domain is the most complex domain studied in the meta-RL setting to date [1, 2].
>
> Independence assumption: when the assumption does not hold, applying reward relabelling amounts to adding tasks to the data that have zero probability under the prior. This should not hurt the general performance of our method, as the VAE would simply learn to identify more possible MDPs. It can, however, make the agent search for the correct MDP less effectively, as it now has more MDP possibilities in the prior. If we were to add such an experiment, it is not clear what the comparison would be, as we do not have a better method for handling MDP ambiguity when the assumption does not hold.
>
> Baselines: we are not aware of stronger baselines for the setting we consider. This is the first study of deep Bayesian RL in the offline setting, and the relevant prior works are all based on Thompson sampling (TS). Therefore, we compared against the strongest TS baselines that we can think of: exact computation in the discrete Gridworld, and online PEARL in continuous domains.
> If the reviewer has a stronger specific baseline in mind, we will happily add comparisons.
>
> Model-based approach: we emphasize that our method uses offline data *only during training*. At test time, it does not require any data from the new task prior to running the learned policy! The model-based RL approach that the reviewer suggests does not answer the question how to explore effectively, which is exactly what our agent learns. To further clarify, observe for example the Ant-Semi-circle task -- here our learned agent finds the goal after only 2 episodes in a new test MDP. The agent's strategy for collecting information is approximately optimal, and any other method, model-based or model-free, cannot discover the reward with less than 2 episodes of interaction data.
>
> Table 1 is now full in the new, updated version.
>
> [1] Rakelly et al. Efficient off-policy meta-reinforcement learning via probabilistic context variables. arXiv preprint arXiv:1903.08254, 2019.
>
> [2] Mitchell et al. Offline meta reinforcement learning with advantage weighting. arXiv preprint arXiv:2008.06043, 2020.

---

### Official Review · AnonReviewer4 · 2020-11-03
**I have concerns regarding assumptions made by the approach and experimental evidence**

**Rating:** 6
**Confidence:** 4

**Review:**

Summary:
This submission studies the meta-learning problem in RL under offline settings. A new algorithm is proposed to address this problem by extending the recent VariBAD algorithm designed for online meta-RL. The key modifications to adapt the original VariBAD to offline settings are the state re-labelling and reward re-labelling tricks, which aim at addressing the MDP ambiguity specifically showing up in offline meta-RL. Experiments are conducted on four sparse reward tasks under both offline and online settings, comparing with PEARL (a recent online approach) and the original VariBAD respectively.

Pros:
1. The problem studied is interesting and important in RL, and largely open.

2. The proposed tricks for adapting VariBAD to offline settings are simple and easy to implement.

3. The ablation study on the Ant-Semi-circle task is helpful.

Cons:
1. Regarding the reward re-labelling trick, the assumption that reward functions for all training environments are known limits the applicability of the proposed algorithm for real-world problems.

2. Regarding the experiments comparing with online PEARL, the training curves of PEARL in Fig. 3 do not seem correct to me, it is strange that PEARL does not make any progress with training, which is very different from the behavior reported in the original PEARL paper.

3. Regarding the soundness of comparison experiments, it would be much more convincing if the following things are taken care of:

 (a) All four tasks reported in the original VariBAD paper are included, rather than just picking one from them.

 (b) Averaging performance is reported over 5 runs rather than just 3, especially considering that SAC could perform quite differently between different random seeds.

 (c) Hyper parameters are set to be more consistent for the same RL backbone (say SAC) across different tasks. It is understandable that two sets of parameters might be needed for online and offline settings.

Based on my main concerns regarding the assumptions on reward function and soundness/correctness of experiments, I lean towards a rejection.

Other questions:

1. Regarding the state re-labelling trick, I have a question regarding the proof of Proposition 1: the first equality of the proof in Appendix A does not seem straightforward to me. It would be helpful if more derivation details are provided.

2. I feel that the claim of approximating Bayes-optimal policy in the offline setting is overly stated. Is there any explicit arguments showing that this is "approximately" true?

====================================================
Post rebuttal:

My concerns regarding the experiments are mostly addressed, though as pointed out by other reviewers, more convincing experiments under changing transition dynamics would be very helpful. I also stand by the authors explanation regarding limited resource in running RL experiments, especially for novel research directions.

That said, for the same reason (pioneering research vs large-scale application), I do not agree with the authors explanation regarding the limitation of the proposed approach in assuming a known ground truth reward function. The main contribution of this submission is not in solving a specific real-world RL application problem as the cited references. As one of the first efforts in addressing the meta-RL under offline setting, I feel that setting this constraint is a limitation and should be relaxed by means of estimating the reward function. This should be addressed in future work.

I raise my rating to a weak acceptance conditioned on the wording regarding "Bayes optimal" being more precisely presented, I think it is currently over-stated throughout the paper, which could be misleading to the community if published as it is.

It is important to carefully reword in which sense the proposed algorithm "approximate Bayes-optimal policy", as explained in the authors' response, the algorithm is shown to qualitatively behave in a way that a Bayes optimal policy would do under this particular setup, this is far from sufficient to claim any approximation to the Bayes-optimality in a principled sense. I would like to also point out that while Bayes-optimality is generally intractable, it is possible and not uncommon for a method to start from an explicit pursue of Bayes optimal solution and specify where and how approximations are performed to overcome specific intractibilities, and further show quantitatively that such Bayes optimality is indeed achievable under well-controlled toy examples where the true Bayes optimal solution is known. The concept of Bayes optimality is in essence quantitative, rather than qualitative.

---

> ### Author Response · Authors · 2020-11-18
> **Reply to reviewer 4**
>
> PEARL performance: In Figure 3 we do not present the training curves of PEARL, but only its *best performance* during online training. We added the training curves of PEARL to Appendix G.2. The rationale for this presentation is that comparing offline and online methods in terms of reward vs. training iterations is rather meaningless.
>
> Known reward function: our method requires knowing the reward function only during training. We are not aware of any real-world RL application in which the reward function was not known ([1, 2, 3]). On the contrary, the reward is typically the practitioner's method of specifying the task!
>
> SAC hyperparameters: some hyperparameters for SAC should be chosen proportional to the action dimensions (Appendix D in [4]), which are different in our domains. Further, it is common in RL to choose different hyperparamters for different domains (see, e.g., PEARL, VariBAD, and SAC (Appendix D in [4])). From the trajectory visualizations, it is clear that the superior performance of our method with respect to PEARL is not due to hyperparameter tweaking, but due to the Bayesian RL formulation.
>
> Random seeds: we will add more seeds to the evaluation.
>
> Evaluation tasks: please see our general comment on this issue above. In particular, we added sparse reward tasks that are much more difficult than the tasks in the VariBAD work.
>
> Proposition 1: we marginalize over possible reward and transition functions and use conditional probability followed by the definition of conditional expectation (see full derivation in new version, page 11).
>
> Bayes-optimality: finding the Bayes-optimal is generally intractable. However, in domains with sparse rewards and uniform prior over MDPs, the Bayes-optimal policy amounts to visiting as many states as possible, without revisiting states, and in case the goal state is found - staying at the goal. Although we do not provide any theoretical guarantees regarding the optimality of our learned policy, we qualitatively show that our trained agent produces trajectories that (approximately) match such Bayes-optimal behavior.
>
>
> [1] Abbeel et al. An application of reinforcement learning to aerobatic helicopter flight. Advances in neural information processing systems, 19:1–8, 2006. \
> [2] Gu et al. Deep reinforcement learning for robotic manipulation with asynchronous off-policy updates. In 2017 IEEE international conference on robotics and automation (ICRA), pp. 3389–3396. IEEE, 2017. \
> [3] Schoettler et al.  Meta-reinforcement learning for robotic industrial insertion tasks. arXiv preprint arXiv:2004.14404,2020. \
> [4] Haarnoja et al. Soft actor-critic algorithms and applications. arXiv preprint arXiv:1812.05905, 2018.

---

### Author Response · Authors · 2020-11-18
**General comment to all reviewers**

We thank all the reviewers for their feedback.

Following several reviewers comments, we would like to clarify the discussion about MDP ambiguity, reward relabelling, sparse rewards, and tasks with uncertainty in the transitions.

The difference between MDPs can be either in the rewards, the transitions, or both. The differences can also take place over the whole state space, or only in parts of it -- i.e., in some states the MDP rewards/transitions are similar, while on others they are different. Let us denote the set of states where there are such differences as the `identifying states' of an MDP -- if the agent has data on such states obtained from different MDPs, it has the capability to identify which data samples belong to which MDP.

The MDP ambiguity problem arises when for each MDP in the data, the samples that belong to identifying states do not overlap. Since the agent does not know the MDPs in advance, it does not have information to identify which MDP the data came from. As we explained in Section 3.3, it may be one MDP that all the different identifying states belong to, or it can be several different MDPs, each with its set of identifying states. Since there's no overlap in the data, there's no way to tell.

Problems with sparse rewards are therefore prone to MDP ambiguity -- the identifying states are the sparse set of rewarding states, and in the way we collect data, each agent mainly visits its own rewarding states. This is the reason that our sparse reward tasks are challenging, and reward relabelling makes a big difference. In the dense Half-Cheetah-Vel domain, on the other hand, almost every state is identifying, and therefore ambiguity is not a problem and reward relabelling doesn't help.

Our approach is not limited to problems with reward uncertainty, and we expect it to work on MDPs with  varying transitions. However, on problems where the MDP ambiguity manifests in the transitions -- that is, the set of states with identifying transitions is sparse, and in the data there is no overlap between the identifying states on different MDPs -- we expect our method to fail. Reviewer 3 gave a great example of such a domain. To our knowledge, previous work on online and offline meta-RL did not investigate such problems. For example, in the Walker environment from [1], the shape of the walker is varied, which manifests in almost every transition.

One way to solve MDP ambiguity in transitions is to add trajectories that run the policy for agent i on task j (this is a generalization of the reward relabelling idea). This, of course, requires collecting more data, and cannot work in the offline setting. It does, however, provide a guideline for effective data collection for offline meta-RL.

We will add the discussion above to the manuscript, and clearly specify the limitations of our approach. We will also add an experiment with a `dense' uncertainty in transitions.


We also want to discuss the comments on our evaluation tasks.
Our offline meta-RL experiments require significantly more resources than their online counterparts. For each task, we first need to successfully train ~100 SAC agents to collect data. Only then we can train our VAE and run offline RL. As our compute resources are limited, we carefully curated a set of experiments that provide insight into our method. This includes a very challenging sparse reward ant (much more challenging than any domain in the original VariBAD work), domains with discrete and continuous states, and tasks for which we have an idea how the Bayes-optimal solution looks like. We believe that our experiments back up our claims (up to the limitations raised by reviewer 3). We will also share our code and data.

We understand that it always helps to add more experiments, and that highly-budgeted research labs have raised the bar for experimental evaluation in RL. That said, we believe there is room for smaller scale studies, so long as the scientific contribution is clear.


[1] Zintgraf et al. Varibad: A very good method for bayes-adaptive deep rl via meta-learning. In International Conference on Learning Representation (ICLR), 2020.

---

### Author Response · Authors · 2020-11-25
**New Experiments + Final General Comments**

Dear reviewers,

Following up on the discussion about MDP ambiguity and variation in rewards and transitions, we have added experiments on offline meta RL with varying transitions, but where MDP ambiguity is not an issue. We have also added a detailed discussion that addresses the points raised in the reviews in Sec 3.3, and clarified (and toned down) our claims.

The new experiments consider a point robot with a fixed goal, but with varying winds that change its transition function. To navigate correctly to the goal and stay there, the agent must choose actions that ‘cancel’ the wind effect. Here, ambiguity is not a problem, as all agents must traverse similar states to the goal, and the transitions on the way are identifying. See Section G.3 for the results. Our learned policy solves this problem much better than the PEARL baseline (mainly because it can identify the MDP during the first episode, and react accordingly, while PEARL only reacts after the end of the episode). This simple demonstration shows that our method can handle varying transitions, so long as ambiguity is not a problem. We modified our claims to clarify the limitations of our approach.

We wish to thank again the reviewers for their insights, which significantly improved the presentation of the problem and our contribution.

---

### Decision · Program_Chairs · 2021-01-07
**Final Decision**

**Decision:**

Reject

**Comment:**

The paper studies offline meta reinforcement learning. Overall the scope of this contribution seems limited. Reviewers have raised concerns about the significance of the presented results given the assumptions, and that the experimental environments are not extensive and do not fully support the claimed advances.